# [Re] What do neural networks learn in image classification? A frequency shortcut perspective

NULL[1, 2] and NULL[1, 2]
[1]Shared authorship – [2]Independent

Edited by
(Editor)

Reviewed by
(Reviewer 1)
(Reviewer 2)

Received
11 November 2023

Published
–

DOI
–

## Reproducibility Summary

**Scope of Reproducibility** – This paper[1] presents an empirical study on the learning dynamics of neural networks for classification tasks through the use of frequency analysis. Our goal is to reproduce experimental results presented by the authors in the paper, which consist of frequency analysis conducted on both synthetic datasets and the ImageNet-10 dataset.

**Methodology** – To reproduce the results, we utilized the paper's authors' existing code from their repository and modified the code for better clarity and code optimizations. In addition, we added additional code for new experiments, implemented new custom Pytorch transformations by their descriptions in the paper, and were able to successfully create a Python package that could be built and installed locally by source. The code was ran locally using several computer architectures, including an NVIDIA 1000 series GPU on Ubuntu 22.04, as well as a 16GB M1 Macbook Pro.

**Results** – We were able to reproduce much of the results from the paper, including the ADCS visualizations, F1-score calculations from training on a synthetic dataset. In addition to this, we also achieved new results by experimenting with a new dataset and introducing new data augmentation techniques. While we were unable to completely replicate the authors' precision and recall visualizations, we recreated those experiments based on our best guesses of the authors' descriptions in the paper. We were able to organize the code used to achieve these results within simple Jupyter notebooks and a Python package. Due to many missing implementation details, specifically with regards to producing their synthetic datasets as well as the lack of clarity on the transformations utilized for their experiments with the ImageNet dataset, we were not able to reproduce all the original results, such as the DFM calculations and relative confusion matrices.

**What was easy** – The training code provided by the paper was, for the most part, well-written and straightforward to use and edit. In the paper, the authors provided specific details regarding the major models used to produce their experimental results. The code as well as the descriptions of the frequency distribution comparison metric ADCS were especially well-presented, and we had almost no trouble producing similar results to the authors for those experiments.

**What was difficult** – Some of the libraries required for the authors' code were incompatable due to new releases, and we had to downgrade some versions. A few dependecies were also not installed in the setup process and had to be added by us. The existing codebase contained many unused functions and hardcoded variable values that had to be modified during training as well. In addition, there was a lack of clarity on the transformations utilized for their experiments with the ImageNet-10 dataset. This was especially troublesome due to the fact that using our own attempted guesses at the transformations utilized, we produced results that were quite different from the original authors'. In addition, the DFM experiments were simply too computationally expensive for us to reproduce in the time frame given.

**Communication with original authors** – An attempt of communication with the first author of the paper is made, though no replies have been received yet as of the submission of the paper.

# 1 Introduction

Though deep neural networks (DNNs) are now being widely used to tackle problems in many fields, the underlying predictive processes of DNNs are still not completely understood due to the DNNs' black-box nature. This issue is becoming especially problematic with the increasingly massive neural network architectures that are being developed and trained[2]. While these large neural networks generally have stronger predictive power than their predecessors, their sheer number of parameters (often in the scale of millions) limit the understanding of the neural network's learning process. The lack of understanding of what happens within the black-box of learning algorithms has led to more insidious problems, such as racial bias within healthcare management algorithms[3], as well as outright tragedies, such as the 2018 accident involving a self-driving Uber car which killed a pedestrian[4]. These "AI accidents" are often caused by undiscovered biases within the trained neural network, and is a major drive behind today's research in explainable AI[5].

In the past, researchers have worked on explaining the predictions of neural networks in terms of their input, using Saliency[6], Gradient-weighted Class Activation Mapping[7], and Layer-wise Relevance Propagation[8]. However, while these techniques highlight areas of the input images that contribute to model predictions, they are still unable to explain the degradation of the performance of neural networks on out-of-distribution data. Therefore, researchers are beginning to look into understanding the learning dynamics of neural networks through frequency data. Prior works has found that neural networks tend to learn lower frequencies first in regression tasks[9]. However, at the time of publication of the original authors' work, there has been little research conducted on the frequency learning behaviour of neural networks in image classification. The authors previously produced a brief paper explaining frequency shortcuts[10], which are the sets of class-based frequency patterns that neural networks may be using to learn to classify images. Frequency shortcuts are dangerous and can lead to a lack of generalization, as frequency shortcut learning can lead to a model learning to categorize images based on a particular color, texture, or shape, instead of more generalizable properties of the class. Therefore, in this new paper with results which we are attempting to replicate, the authors conducted an empirical study on the learning dynamics of neural networks for image classification. The authors relate their observations to simplicity bias and short-cut learning, biased behaviours commonly seen in the training of deep neural networks[1].

## 2 Scope of reproducibility

The main goal of this report is to reproduce the experimental results that were mentioned within the paper by Wang et al.[1] in order to investigate the reproducibility of the paper as well as to verify the claims made by the paper's authors. The paper conducts experiments on a custom synthetic dataset, as well as the commonly used ImageNet-10 dataset and its various data augmented variations. An overview of the experiments, grouped by the dataset on which they are conducted, is presented as follows:

**Experiments on Synthetic Data:**

- F1-scores by class for the first 500 training iterations of the models AlexNet, ResNet18, and VGG16.

- Relative confusion matrices of AlexNet and VGG16 trained on the four synthetic datasets and tested on the different band-stop test sets.

- ADCS computation and visualization of the four synthetic datasets by class.

**Experiments on ImageNet10:**

- ADCS computation and visualization of the ImageNet-10 dataset by class.

- Precision and recall scores by class for the first 1200 training iterations of the models ResNet18, ResNet50, and VGG16, computed on the low-passed and high-passed test sets of ImageNet-10.

- Computation of the top-1% and top-10% DFMs of each class for models trained on ImageNet-10.

- Classification results of models tested on ImageNet-10 DFM-filtered versions, with only the top-1% and top-10% dominant frequencies retained.

In the end, we were able to complete all experiments related to the computation of ADCS, and were able to successfully complete the computation of F1-scores by class for the first 500 training iterations of the different models utilized by the authors of the paper. However, we had problems replicating the authors' results for their precision and recall scores when training the models on ImageNet-10 for the first 1200 iterations, and were unable to compete DFM computation due to computational constraints.

## 3 Methodology

### 3.1 Datasets

**Synthetic Datasets –** The authors of the original paper[1] created synthetic datasets to test how frequency shortcuts affect neural networks in a direct way. Four main datasets were created, and each contains a frequency bias in a different band based on the Fourier spectrum. The dataset $B_1$ is biased toward the lowest band, the datasets $B_2$ and $B_3$ towards mid range bands, and the dataset $B_4$ towards the highest band. Each dataset consists of 4 classes and is made up of $32 \times 32$ images. The classes are structured so that $C_3$ is made up of images with the same frequency band that the respective dataset is biased towards, which makes it the simplest in terms of frequency. $C_0$ contains special patterns that are removed from other classes, which also makes it simpler in terms of frequency than classes $C_1$ and $C_2$. Based on the idea that neural networks are prone to a simplicity bias[11], the authors hypothesized that the classes $C_0$ and $C_3$ would be easier for the networks to learn.

**ImageNet10 –** ImageNet-10[12], which is a smaller version of the popular ImageNet[13] dataset, is a dataset containing $224 \times 224$ images featuring 10 classes of every day objects such as wagons, zebras, and trucks. The relatively smaller ImageNet-10 was chosen as opposed to the full version in order to reduce computation time, simplify the analysis process, and to allow for a more thorough analysis of the dataset. This dataset was used to examine how frequency shortcuts affect the way neural networks learn using an approximate set of real-world data.

## 3.2 Models

**ResNet –** ResNets[14] are a class of popular convolutional neural network architectures that use residual nets. Various versions of ResNet, such as ResNet18 and ResNet50 were used in the paper. ResNet18 was also tested with three different common data augmentation methods, AutoAugment[15], AugMix[16], and SIN[17]. These augmentation methods were tested by the authors to determine how effective data augmentation techniques are at avoiding frequency shortcuts in neural network training.

**VGG-16 –** VGG-16[18] is another popular convolutional neural network architecture that has shown to be effective at classifying images from the ImageNet dataset.

**AlexNet –** AlexNet[19] is a large, deep convolutional neural network that was used in addition to ResNets and VGG-16 to test and examine the effects of frequency shortcut learning on synthetic and real datasets.

## 3.3 Hyperparameters

Following the directions of the original paper, an initial learning rate of 0.01 was used for all model training, and the learning rate was reduced by a factor of 10 if the validation loss score did not improve after 10 epochs. Batch sizes of 64, 32, and 16 were used depending on the model and dataset, as larger batch sizes would lead to our GPUs running out of memory and crashing during training. For training where the F1-scores, precision, or recall were tracked, the termination condition was set to when the training hit either 500 or 1,200 steps. For regular training, the termination condition was set to a max number of 100 epochs when training on Synthetic datasets, and 200 epochs when training on the ImageNet-10 dataset. More details involving our specific hyperparameters for various training configuration are saved in config files in our repository.

## 3.4 Metrics

**Accumulative Difference of Class-wise average Spectrum (ADCS) –** The ADCS (or Accumulative Difference of Class-wise average Spectrum) is a metric devised by the authors of the paper, which is utilized to examine the frequency characteristics of individual classes within a dataset[1]. The metric computes the average amplitude spectrum difference per channel for each unique class within a dataset, and then averages it into a one-channel ADCS. For a class $c_i$ in the set of classes $C = \{c_0, c_1, \ldots, c_n\}$, the ADCS for $c_i$ at a frequency $(u, v)$ is computed as

$$\text{ADCS}_{c_i}(u, v) = \sum_{\forall c_j \neq c_i \in C} \text{sign}[E_{c_i}(u, v) - E_{c_j}(u, v)]$$

where $E_{c_i}(u, v)$ is the average Fourier spectrum for class $c_i$ and is computed as

$$E_{c_i}(u, v) = \frac{1}{|X_i|} \sum_{x \in X_i} |\mathcal{F}_x(u, v)|$$

In the above formula, $x$ denotes an image from a set $X_i$ of images that are contained in the class $c_i$, and $\mathcal{F}_x(u, v)$ is the Fourier transform of the image.

The ADCS metric is useful for analysing the class-based frequency patterns that exist in a dataset, and it can be used to show how neural networks may become biased by identifying these classes based on their respective frequency trends during training. The ADCS metric ranges from $1 - |C|$ to $|C| - 1$, where $|C|$ is the number of unique classes within a dataset. A higher value for a specific frequency represents that the class has more energy at that frequency than other classes, and a lower value represents it has lower energy.

$F_1$-**Score –** For experiments on the early training stages of neural networks which were trained on the four Synthetic datasets, $F_1$-scores are calculated at each step. The $F_1$-scores are calculated by class in order to observe differences in the neural network training progress when different classes display different frequency patterns, which were discovered via the computation of ADCS. The exact $F_1$-score calculation is

$$F_1 = \frac{2 \times \text{precision} \times \text{recall}}{\text{precision} + \text{recall}}$$

It is a highly useful tool for getting a single metric that represents how accurate a model is at classifying data of a certain class.

**Precision and Recall –** For experiments on the early training stages of neural networks which were trained on the low-pass and high-pass versions of the ImageNet-10 dataset, precision and recall are calculated at each step. The precision and recall metrics are similarly calculated by class in order to observe differences in the neural network training progress when different classes display different frequency patterns, which were discovered via the computation of ADCS.

**Relative Confusion Matrix –** Relative confusion matrices were used to compare the classification results of neural networks tested on original datasets and their band-stop versions in order to examine how neural networks find shortcuts in their training process[1]. The relative confusion matrix is calculated as

$$\Delta(c_i, c_j) = \frac{\text{Pred}_{\text{bs}}(c_i, c_j) - \text{Pred}_{\text{orig}}(c_i, c_j)}{N(c_i)} \times 100$$

where $\text{Pred}_{\text{bs}}(c_i, c_j)$ and $\text{Pred}_{\text{orig}}(c_i, c_j)$ are the number of samples from class $c_i$ that are predicted by the model to be from class $c_j$, using the band-stopped test set and the original test set respectively, and $N(c_i)$ is the number of samples in class $c_i$. When $\Delta(c_i, c_j) \geq 0$, this indicates that the performance of the model improves or remains the same on the band-stopped test sets, and when $\Delta(c_i, c_j) < 0$, this indicates that the performance of the model decreases for the band-stopped test sets.

**Top-$X$% Frequencies and DFMs –** Dominant-frequency maps, or DFMs, are computed using the ranked top-$X$% important frequencies for classification[1]. The top-$X$% important frequencies were ranked by utilizing the change in loss value when testing a model on images of a certain class with the concerned frequency removed from all channels. Then, the DFMs are utilized to filter the original dataset, and the true positive rate (TPR) and false positive rate (FPR) metrics are calculated to evaluate the discrimination power of those frequencies for different classes. Those classes that are found to have higher TPR and FPR are those that the neural network model tends to learn and apply frequency shortcuts in its training progress.

# 4 Results

## 4.1 Reproduction of ADCS Results Using Synthetic Datasets

For our replication, we computed graphics that visualize the ADCS of the classes within the four full synthetic datasets (i.e., no filtering of frequency data by fourier spectrum bands). The computed graphics are presented below:

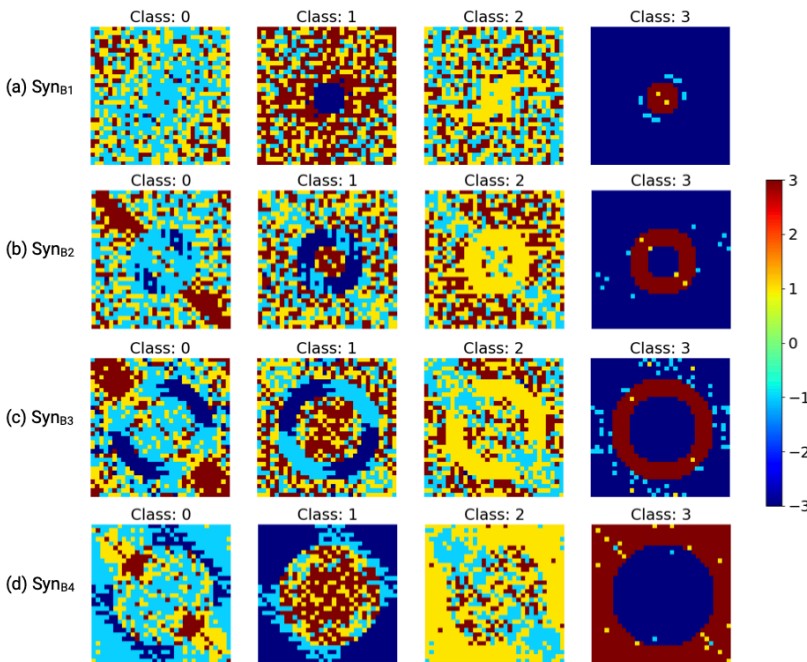

**Figure 1**. ADCS heatmaps computed for synthetic datasets by class: lower values (blue) indicate that the class has lower energy for those frequencies while higher values (red) indicate that the class has higher energy for those frequencies.

Based on the graphics published within the supplemental data of the original paper, we can see that we have obtained similar results to the authors, though the results are not the exact same, despite the fact that the computation of ADCS should be deterministic. It should also be noted that we have used the Synthetic dataset that was provided by the authors rather than generating our own version of the Synthetic dataset based on the descriptions of the authors in the paper. This indicates that there could either be some discrepancies in the synthetic dataset provided by the author, or that there are missing implementation details for the computation of ADCS values using the synthetic datasets. The figure produced in the original paper is presented below for reference and comparison.

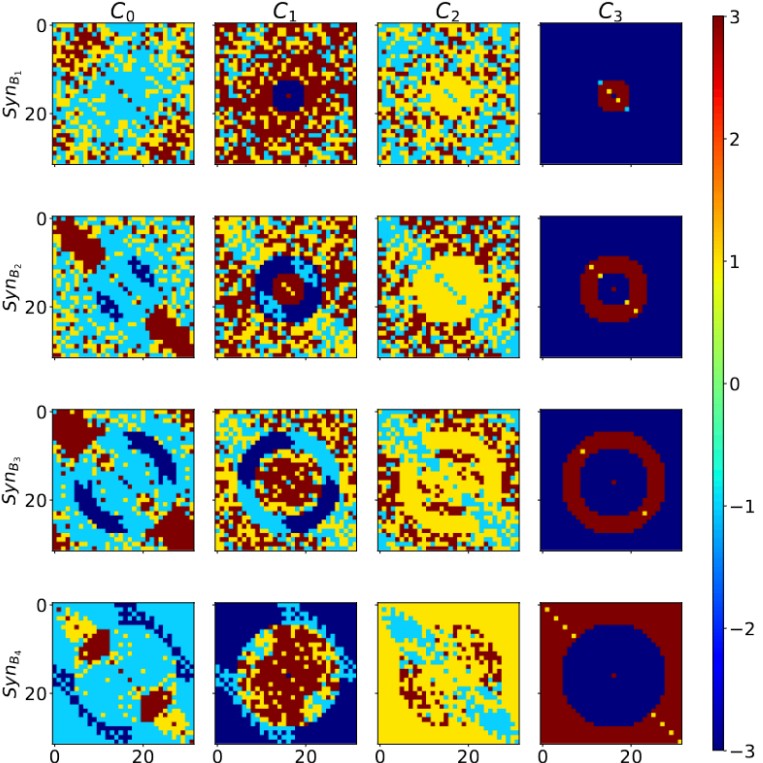

**Figure 2.** Original ADCS heatmaps computed for synthetic datasets: it could be seen that there are discrepancies in our replicated results from the original paper, such as more higher energies detected for Class $C_3$ in our results in comparison to the original authors'.

While the authors provided code for ADCS computation, the code as was provided had hardcoded values designed for ADCS computation on the ImageNet-10 dataset. Therefore, we extensively refactored the original code and reformatted it as part of our custom Python package `nnfreq`. The reformatted code is part of the subpackages `nnfreq.adcs` and `nnfreq.data`. In addition, we created a Jupyter notebook which produces all the experimental results that are documented in this paper.

## 4.2 Reproduction of ADCS Results Using ImageNet-10

Similar to our reproduction of ADCS results using synthetic datasets, we computed graphics visualizing the ADCS of the classes within the ImageNet-10 dataset to see how our results compare to the ADCS visualizations from the same dataset in the original paper. The computed graphics for this experiment are presented below:

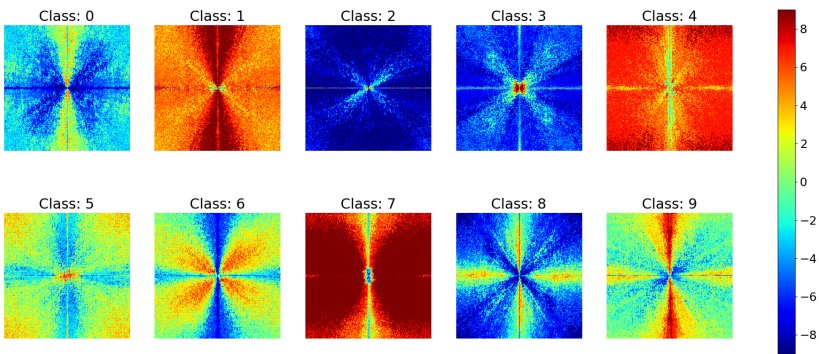

**Figure 3**. ADCS computed for the original ImageNet-10 dataset.

The original graphics from the paper are also presented below for further reference and comparison.

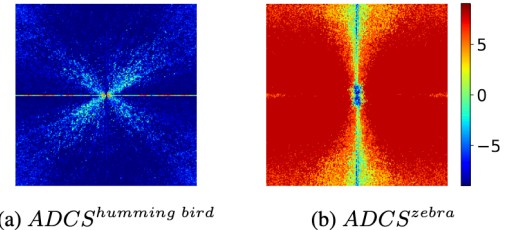

(a) $ADCS^{humming\ bird}$          (b) $ADCS^{zebra}$

**(a)** Original ADCS heatmaps computed for the "humming bird" and "zebra" classes within the original ImageNet-10 dataset (class 2 and class 7 respectively).

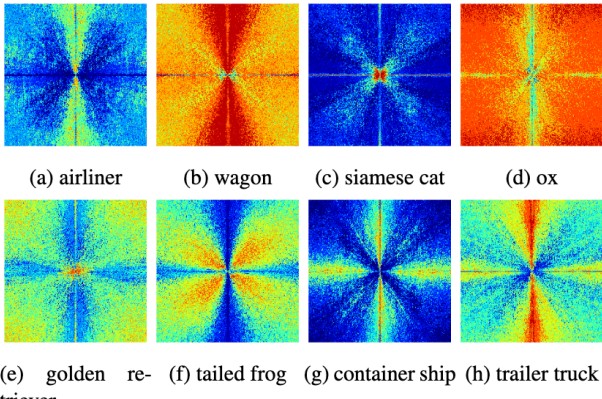

(a) airliner          (b) wagon          (c) siamese cat          (d) ox

(e) golden re-trieve          (f) tailed frog          (g) container ship          (h) trailer truck

**(b)** Original ADCS heatmaps computed for all other classes within the original ImageNet-10 dataset.

**Figure 4**. Original heatmaps computed for the original ImageNet-10 dataset.

Overall, we achieved similar results when recomputing the ADCS of the ImageNet-10 dataset. Within the original paper, the author's claim that the classes "humming bird" and "zebra" posess distinctive frequency characteristics that can be readily exploited by models to distinguish them from other classes at early training stages[1]. This claim can be verified based on our replicated results presented in Figure 3, where classes 2 (represents class "humming bird") and 7 (represents class "zebra") appear to have energy more biased to either the lower end or higher end for all frequencies.

In addition to the results we replicated from the original paper, we also created a new experiment by computing an ADCS visualization on Stylized-ImageNet[17], which is a modified dataset based on ImageNet that contains stylized versions of the original images. Compared to the results of the ImageNet10 dataset, the frequency patterns are not as clearly defined. This suggests that using the Stylized-ImageNet-10 dataset rather than the original ImageNet-10 dataset to train deep neural networks would decrease the neural networks' ability to learn shortcuts in the early stages of the training process. However, there are still existing frequential patterns, which suggests that the data augmentations performed on the original ImageNet-10 dataset can not eliminate all frequency shortcut learning.

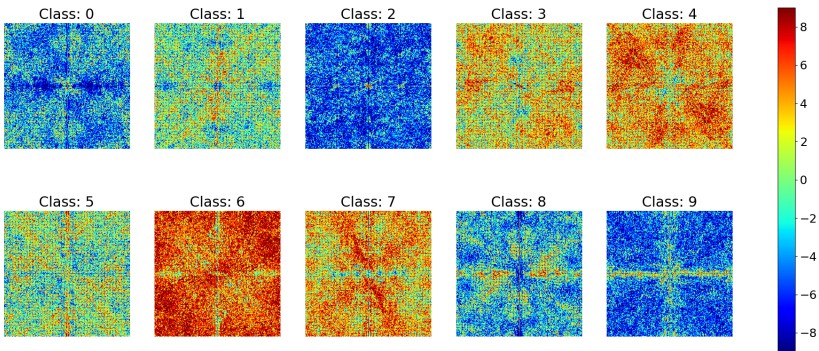

**Figure 5.** ADCS computed for Stylized-ImageNet

## 4.3 Reproduction of F1-Score Heatmaps Using Synthetic Datasets

The authors of the original paper recorded the $F_1$-scores of various models during their training on the synthetic datasets. These $F_1$-scores were logged for the first 500 steps of training, and they were recorded by class. As discussed in the synthetic dataset section of this paper, the authors expected that all of the models would better categorize images in classes $C_3$ and $C_0$, as images from those classes are simpler in terms of frequential analysis. We were able to recreate the original paper's experiments on all of the models and synthetic datasets they tested, which includes ResNet18, VGG-16, and AlexNet trained on the four synthetic datasets. Our results differed due to variations in step size and random training variation. However, we discovered the same trends as the original paper, and we reached the same conclusion that $C_3$ and $C_0$ are easier to classify than the other synthetic classes. The results for AlexNet are shown below, and the results for ResNet18 and VGG-16 showed a similar class-based pattern.

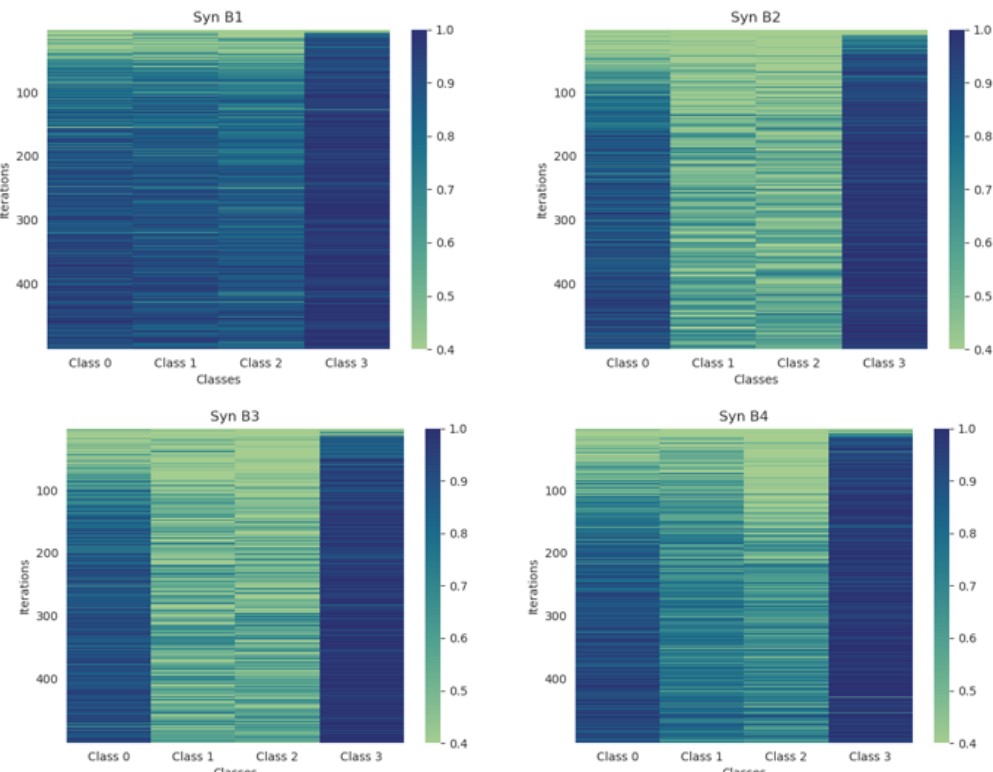

**Figure 6.** $F_1$ scores computed by class for the first 500 steps of AlexNet trained on the four synthetic datasets. It is clear visually that the model classifies $C_3$ most effectively and $C_0$ second most effectively.

## 4.4 Reproduction of Precision and Recall Heatmaps Using ImageNet-10

The authors of the original paper also stated that they recorded the precision and recall scores of various models for the first 1200 steps of the training on high-pass and low-pass versions of the ImageNet-10 dataset. Unfortunately, we were not able to find the code they used to create high-pass and low-pass versions of original ImageNet-10 dataset, and descriptions in the paper were unclear on that front. Therefore, we made our best guess with the implementation of high-pass and low-pass filtering.

For our low-pass filter implementation, we created images with a black background and a white circle in the foreground, where the size of the circle is a parameter that can be passed as an argument, either in the form of a percentage of the size of the image or a fixed radius. Then, a fourier transform is applied to each channel of the input image, and the output results are multiplied with the filter. Finally, an inverse fourier transform is applied to each channel of the product to regenerate the filtered image. The same process is followed to create high-passed input images, except in the last step, we also subtract the low passed image from the original image to obtain our final output. The algorithm used to implement these filters could be found in the subpackage `nnfreq.transforms`.

However, our produced results differ quite differently from the original authors' results. Our results are presented below, followed by the original authors' results:

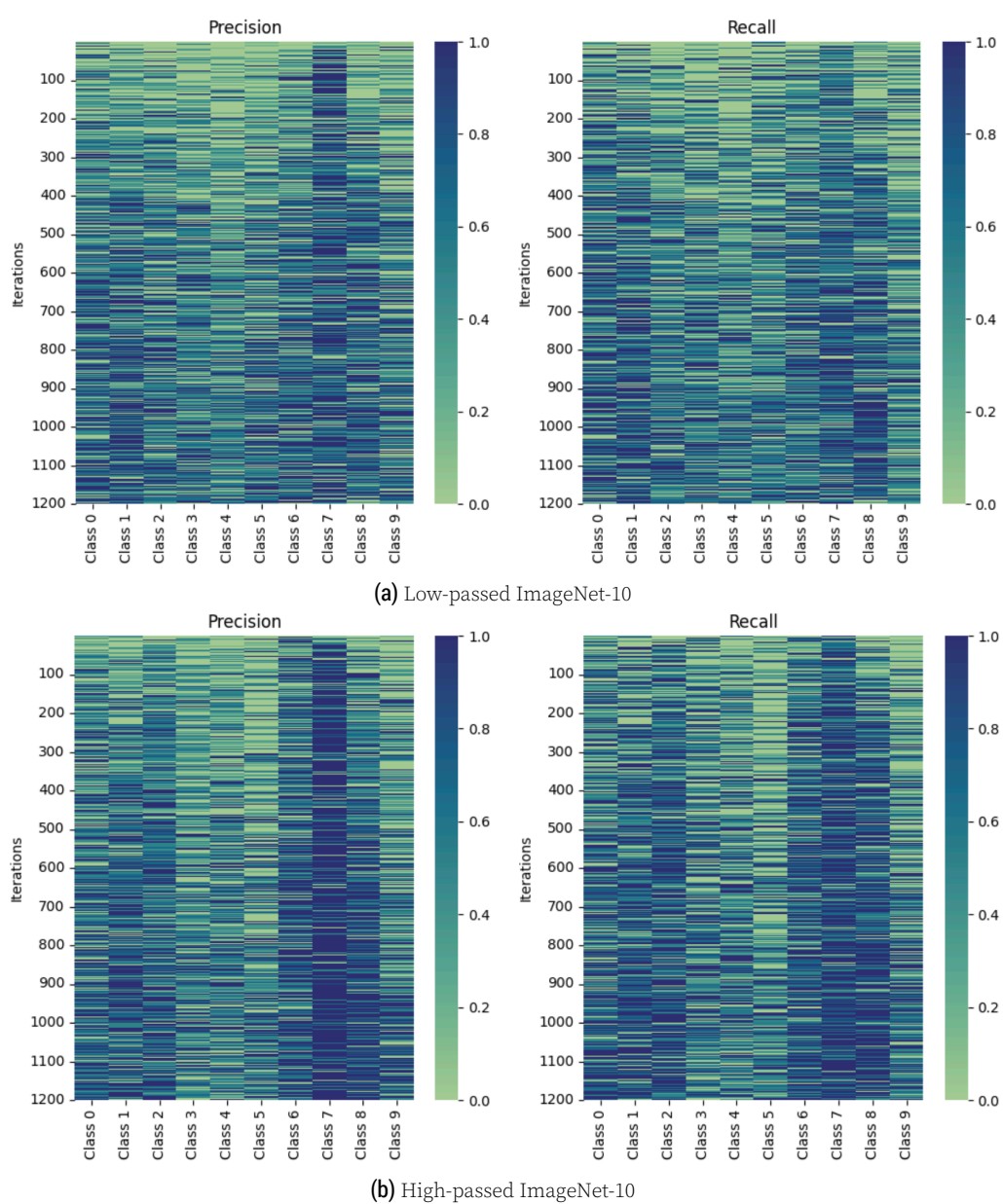

**Figure 7**. Precision and recall scores of the first 1200 training iterations of ResNet18 on the low-passed and high-passed versions of the ImageNet-10 dataset.

It is clear that our results presented in figure 7 are quite different from the results presented by the authors in the original paper, which we have also presented below in figure 8. Based on the labelling on the original authors' graphics, it is likely that the discrepancy is due to the fact that the original authors have used a band-pass filter rather than a low-pass and high-pass filter separately. However, further correspondence is likely needed with the authors in order to confirm the exact nature of the filters they have used.

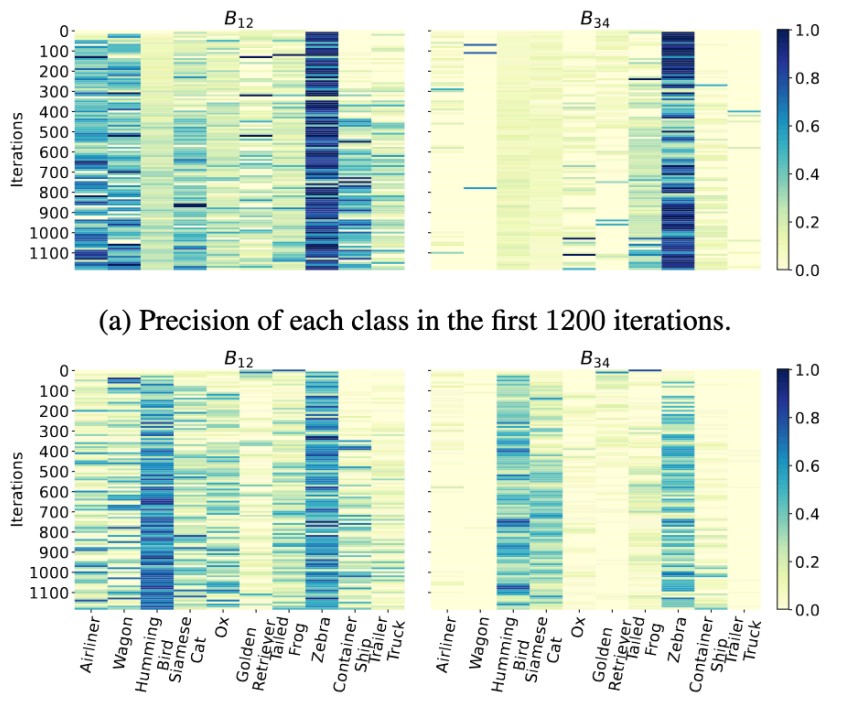

(a) Precision of each class in the first 1200 iterations.

(b) Recall of each class in the first 1200 iterations).

**Figure 8**. Precision and recall scores computed by class for the first 1200 steps of ResNet18 trained on the high-pass and low-pass versions of the ImageNet-10 dataset from the original paper.

## 5 Conclusion

Our experiments generally support the claims and results from the original paper. Through our analysis of the ADCS metric for various classes, our results support the authors' claims that certain classes within a dataset can contain frequency biases, which can lead to neural networks finding frequency shortcuts that they then use to classify images. Based on a visual inspection of our ADCS visualzations, it is clear that datasets, such as ImageNet-10 and Stylized-ImageNet, contain class-based frequency patterns. Our new experiments on Stylized-ImageNet showed certain transformation and manipulations of datasets can help reduce frequency patterns, but they cannot fully prevent the patterns.

Similarly, our results in reproducing the $F_1$-scores for various deep neural networks trained on the synthetic frequency-based datasets supports the authors' claims that neural networks will learn to recognize frequency biases in classes. As classes with simpler, more defined frequency patterns were more quickly picked up by neural networks and classified more accurately, this shows that the models were likely using frequency shortcuts to categorize images, which leads to a lack of generalization. Unfortunately, due to lack of clarity on the exact nature of the filter they have used for computing the precision and recall scores of the models trained on ImageNet-10 in the first 1200 iterations, as of right now, it cannot be concluded that our results support the same claim. However, it is likely that utilizing the same filters as the original authors will enable us to reach similar results.

In conclusion, our replication shows that frequency shortcuts can occur from training on a wide array of datasets, and they are likely leading to misconceptions about how

effective models actually are at generalizing. We conclude by reinstating the original authors' reccomendation that further research is done on the topic in order to better understand how frequency shortcuts are affecting deep neural networks, and further research needs to be performed to determine better ways to avoid frequency shortcut learning.

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
