# OpenReview forum: "[Re] What do neural networks learn in image classification? A frequency shortcut perspective"
_purdue.edu/Purdue_University/ML/2023/Hackathon_Reproducibility_Challenge — Purdue University ML 2023 Hackathon Reproducibility Challenge Submission_

### Official Review · Reviewer_iG2V · 2023-11-27
**Review of Reproducibility Paper: What do neural networks learn in image classification? A frequency shortcut perspective**

**Rating:** 8
**Confidence:** 4

**Review:**

The authors reviewed paper that made the following main claims:
NN’s tend to find simple solutions for classification, and what they learn first during training depends on the most distinctive frequency characteristics, which can be either low- or high-frequencies
Confirming their hypothesis on natural images, the authors proposed a way to identify frequency shortcuts and their results showed that these shortcuts can either be texture based on shape based
The author’s results suggest that frequency shortcuts can be transferred across datasets and cannot be fully avoided by larger model capacity and data augmentation
The review of this paper was very well organized paper, detailing:
Clear introduction and motivation
In depth ML methodology such as
Data set used along with
Training Methodology and Hyperparameters
Models used
Evaluation metrics
Missing
Average runtime
Computing infra used
Analysis of Gaps / Limitation
Results Reproduction
 Claim #1: F1 Score and Precision / Recall results
F1 Score Table substantiates Claim #1
Missing table for VGG19 and ResNet
Precision and Recall tables substantiates Claim #1
Differ probably due to implementation of band pass filter and not low / high pass filter
Claim #2: ACDS results
ACDS results using synthetic dataset
The authors should include details on whether the synthetic dataset is exactly the same as the one used by authors. For ex: # of rows in each class, etc
ACDS results using Image Net dataset
ACDS results on Stylized Image Net dataset: Good results substantiating the author’s claim #3 especially with respect to frequency shortcuts around shape based learning.
Claim #3: Unable to reproduce DFM results in the paper that helped substantiate claim #3.